# OpenReview forum: "You Didn't Have to Say It like That:  Subliminal Learning from Faithful Paraphrases"
_EurIPS.cc/2025/Workshop/UPLB — Submitted to UPLB2025_

### Official Review · Reviewer_sXHt · 2025-10-25
**Not clear contribution**

**Rating:** 3
**Confidence:** 3

**Review:**

The paper focuses on the study of the transmission of preferences/biases to a student that is fine-tuned on different kind of paraphrased sentences generated by a suitably prompted teacher. The goal is to evaluate if/how much the student changes its preferences in light of the bias in the training dataset. The analysis is centered around natural language inputs and outputs.

From the presentation point of view, my personal impression is that, unfortunately, the manuscript fails to achieve a sufficient level of clarity. Since the abstract itself, the text refers to *dolphin- or eagle-loving teachers* and it is not clear what the authors are referring to; in general results are anticipated in Section 1 without providing enough information, so that sometimes concepts appear for the first time in a quite unexpected and confusing way. Just to give an example, at line 47 the authors say ``We find successful transmission for dolphin and eagle preferences, but not for owl, dog, or fly``: up to that point no reference to these animals were given in the text (except for the abstract) so that the reader is left with the task to actually understand what the authors are talking about and why the difference between the result for ``dolphins`` and ``flies`` should matter. The text is quite sketchy (more like work-notes), abbreviation are introduced withouth proper definition although standard (e.g., LLMs or CoT) and data themselves are sometimes dropped without much of a comment (just to give an example, what ```+21pp``` means in line 55 is left to the reader's intuition).  Finally, although I understand that it is not compulsory to have an extensive bibliography, only 6 references were given, which is a little bit surprising given the relevance of the topic.

From the methodological point of view, I have the impression that there is no much control on the dataset features, and this can be problematic. A first dataset is created by using *Claude 4.5 Sonnet* and then fed in *GPT-4.1 nano* to create paraphrases of the original sentences: the quality of the result is checked by another LLM, *GPT-4o mini*, whose ratings allow for a further filtering of the dataset. The entire process involves a number of 'black boxes'. The trust in the quality of such dataset relies on a  'leap of faith' in the fact what LLMs are doing in the generation and in the final LLM 'quality evalution', besides an unclear control of possible mixed bias effects in all these steps. In general, *quantifying* the properties and quality of the input datasets seems dire: I have the impression that the authors are conscious about this, see for example the *Keywords analysis* section, where some interpretations really look far fetched to me (eg, ``the increased use of "<" and ">", which could be interpreted as subtle references to beaks``) and in fact it is admitted that the interpretation of word frequency/pattern is unclear. The role of the training dataset construction and the training baseline might be the reason behind the inconsistency of the results between different types of animals. As the authors said, the general positivity bias of the models might also be another important, uncontrolled ingredient.

In a sense, the results seem to be weaker than the ones in Cloud et al., precisely because the student is fed by data with semantic meaning (as the authors themselves state), and therefore one might expect the finding of Cloud et al. to be, if anything, ultimately amplified.

---

### Decision · Program_Chairs · 2025-11-03

Reject